# Effect of High Hydrostatic Pressure Processing and Holder Pasteurization of Human Milk on Inactivation of Human Coronavirus 229E and Hepatitis E Virus

**DOI:** 10.3390/v15071571

**Published:** 2023-07-18

**Authors:** Peggy Bouquet, Virginie Alexandre, Marie De Lamballerie, Delphine Ley, Jean Lesage, Anne Goffard, Laurence Cocquerel

**Affiliations:** 1Unit of Clinical Microbiology, Institut Pasteur de Lille, F-59000 Lille, France; peggy.bouquet@pasteur-lille.fr (P.B.); anne.goffard@univ-lille.fr (A.G.); 2Univ. Lille, CNRS, Inserm, CHU Lille, Institut Pasteur de Lille, U1019—UMR 9017—CIIL—Center for Infection and Immunity of Lille, F-59000 Lille, France; virginie.alexandre@pasteur-lille.fr; 3GEPEA, UMR CNRS 6144, ONIRIS CS 82225, F-44322 Nantes, France; marie.de-lamballerie@oniris-nantes.fr; 4CHU Lille, Division of Gastroenterology Hepatology and Nutrition, Department of Paediatrics, Jeanne de Flandre Children’s Hospital, F-59000 Lille, France; delphine.ley@chru-lille.fr; 5Univ. Lille, Inserm, CHU Lille, U1286—INFINITE—Institute for Translational Research in Inflammation, F-59000 Lille, France; jean.lesage@univ-lille.fr

**Keywords:** human milk, high hydrostatic pressure, Holder pasteurization, coronavirus, hepatitis E virus, infectivity

## Abstract

In preterm infants, sterilized donor milk (DM) is frequently used for feeding when breast milk is lacking. Most human milk banks use the Holder pasteurization method (HoP) to ensure the microbiological safety of DM. However, this method degrades many bioactive factors and hormones. Recently, high hydrostatic pressure (HHP) processing, which preserves bioactive factors in human milk, has been proposed as an alternative method to ensure the safety of DM. Although HHP treatment has been shown to be effective for viral inactivation, the effect of HHP on viruses that may be present in the complex nutritional matrix of human milk has not yet been defined. In the present study, we compared the efficacy of two HHP protocols (4 cycles at 350 MPa at 38 °C designated as 4xHP350 treatment, and 1 cycle at 600 MPa at 20 °C designated as 1xHP600 treatment) with the HoP method on artificially virus-infected DM. For this purpose, we used human coronavirus 229E (HCoV-229E) and hepatitis E virus (HEV) as surrogate models for enveloped and non-enveloped viruses. Our results showed that HCoV-229E is inactivated by HHP and HoP treatment. In particular, the 4xHP350 protocol is highly effective in inactivating HCoV-229E. However, our results demonstrated a matrix effect of human milk on HCoV-229E inactivation. Furthermore, we demonstrated that HEV is stable to moderate pressure HHP treatment, but the milk matrix does not protect it from inactivation by the high-pressure HHP treatment of 600 MPa. Importantly, the complex nutritional matrix of human milk protects HEV from inactivation by HoP treatment. In conclusion, we demonstrated that HHP and HoP treatments do not lead to complete inactivation of both surrogate virus models, indicating that these treatments cannot guarantee total viral safety of donor milk.

## 1. Introduction

Mothers of preterm infants are frequently unable to provide breast milk in sufficient amounts due mainly to a foreshortened period of preparatory lactogenesis. Human milk banks (HMBs) provide donor milk (DM) as an alternative for the feeding of these infants. To ensure the microbial safety of DM, most HMBs sterilize human milk using the standard method of Holder pasteurization (HoP) performed by heating milk at 62.5 °C for 30 min [1,2,3]. However, this method degrades numerous heat-sensitive bioactive factors, such as immunoglobulins, lactoferrin, some vitamins, lysozyme, the bile salt-dependent lipase (BSSL), and some hormones [1,2,3]. High hydrostatic pressure (HHP) processing is used in the food industry to achieve microbial decontamination of foods for up to 30 years [4]. Recent evidence has demonstrated that HHP may be an innovative method to sterilize DM and, in addition, that this method maintains numerous bioactive factors, such as immunoglobulins, lactoferrin, lysozyme, BSSL enzyme, milk oligosaccharides, and several hormones at levels close to raw milk [5,6,7,8]. More precisely, it was shown that all of these breast milk factors are remarkably well preserved using a HHP protocol of four cycles of a moderate pressure (350 MPa) of 5 min each, at 38 °C [6]. It would, therefore, be attractive to propose HHP treatment for milk sterilization in HMBs. However, although HHP processing has been shown to be effective in viral inactivation [9,10], the effect of this process on viruses that may be present in the complex nutritional matrix of DM remains to be defined. In the present study, to evaluate the efficacy of HHP processing on viral inactivation in DM, we artificially infected pools of milk with two types of viruses that are very different in terms of viral organization: The human coronavirus (HCoV) 229E and the hepatitis E virus (HEV). 

Coronaviruses belong to the *Coronaviridae* family in the order of Nidovirales [11]. Two genera, alpha- and betacoronavirus, infect mammals, among them humans. Since the emergence of SARS-CoV-2, seven coronaviruses infect humans. HCoV-229E and HCoV-NL63 are alphacoronavirus. HCoV-OC43, HKU1, SARS-CoV, MERS-CoV, and SARS-CoV-2 belong to the betacoronavirus genus but are grouped in several subgenera (https://ictv.global/report_9th/RNApos/Nidovirales/Coronaviridae (accessed on 13 July 2023). Coronaviruses are RNA-enveloped viruses. They are enclosed with a lipid bilayer in which viral proteins and glycoproteins are embedded and are essential for viral infectivity. Indeed, three viral proteins are anchored in the lipidic envelope of coronaviruses: The spike protein (S), the membrane protein (M), and the small envelope protein (E) [12]. In the present study, we used HCoV-229E, that is an endemic coronavirus involved in low-pathogenic respiratory tract infections in immunocompetent humans [13]. It replicates readily in cell culture and does not require a biosafety laboratory level-3 (BSL-3) to handle. Therefore, it can be used as a model for studying coronaviruses. 

Hepatitis E virus (HEV), a member of the *Hepeviridae* family [11], is the most common cause of acute viral hepatitis worldwide. Although HEV causes a mostly asymptomatic self-limited disease, HEV infection can lead to chronic disease in immunosuppressed patients or to fulminant liver failure, particularly in pregnant women [14]. During pregnancy, HEV transmission can be transplacental, increasing the risk of abortions and stillbirths and of liver necrosis and deaths in newborns [15,16]. HEV is mainly transmitted enterically by drinking contaminated water or eating raw or undercooked infected meat. HEV is a non-enveloped virus of 27–34 nm in diameter. The 7.2-kb RNA genome encodes three open reading frames (ORF) which are translated into three proteins, including the ORF2 capsid protein [17]. Like hepatitis A virus (HAV) particles, HEV virions in the environment and feces of infected patients are non-enveloped (neHEV), while those in patient blood and urine and culture supernatants are quasi-enveloped (eHEV), i.e., they are enveloped by a lipid membrane but do not express a virus-encoded protein on their surface [18], thus distinguishing them from conventional enveloped viruses, such as coronaviruses.

Although the presence of HCoV-229E and HEV in human milk is poorly documented, these two viruses represent good surrogate models for enveloped and non-enveloped viruses. In the present study, we sought to investigate the effect of two HHP protocols, using different pressures and application times, on DM artificially infected with HCoV-229E and HEV, used as enveloped and non-enveloped virus models, respectively. In parallel, we also evaluated viral inactivation upon Holder pasteurization.

## 2. Materials and Methods

### 2.1. Human Milk Samples, HHP and HoP Treatments

Frozen DM samples from 8 donors were provided by the regional HMB (Lactarium Régional de Lille, CHU Lille). Written informed consent of each donor mother for the use of their breast milk for research studies was obtained by our HMB and validated by the Lille Hospital (authorization number DOC/LAC/009-2012, Lactarium Régional de Lille, Hôpital Jeanne de Flandre). After thawing the individual milk samples, batches of DM were prepared by mixing the individual DM samples to obtain DM of homogeneous composition. Protocols of HHP and HoP applied to DM are described in Figure 1. Viruses were kept undiluted or diluted to 1:2 in culture medium or in DM. Four aliquots of samples were prepared from each batch: One fraction did not undergo any treatment (CTL), one fraction was subjected to 4 cycles of 5 min of a 350 MPa pressure at 38 °C, as previously described [6,7], one fraction was subjected to 1 cycle of HHP of 600 MPa at 20 °C during 5 min, the last fraction was subjected to HoP according to the standard protocol (pasteurization at 62.5 °C for 30 min). After HHP and HoP processing, samples were stored at −80 °C until analysis. Viral inactivation was evaluated by quantification of infectious particles and by quantification of viral RNAs in infected cells, as described below.

### 2.2. Cells and Plasmids

PLC3, Huh-7, and Huh-7.5 [19] cells were cultured in Dulbecco’s modified Eagle’s medium (DMEM, ThermoFisher Scientific, Waltham, MA, USA) containing GlutaMAX-I, 10% inactivated fetal calf serum and 1% non-essential aa (ThermoFisher Scientific, Waltham, MA, USA). PLC3 cells are a subclone of the PLC/PRF/5 (CRL-8024) hepatoma cells and were characterized as highly replicative and productive cell line for HEV [20]. PLC3 cells were authenticated by STR profiling, Huh-7 and Huh-7.5 cells were authenticated by Multiplex Cell Authentication (Multiplexion, Friedrichshafen, Germany). 

The plasmid pBlueScript SK(+) carrying the DNA of the full-length HEV genome of gt3 Kernow C-1 p6 strain (GenBank accession number JQ679013, kindly provided by S.U Emerson) was used.

### 2.3. Virus Production 

The HCoV-229E VR-740 strain (ATCC) was used. It was amplified on Huh-7 cells (m.o.i. = 0.01) cultured in DMEM supplemented with 2% fetal calf serum and incubated for 3–4 days at 33 °C 5% CO_2_. The supernatant containing the viral particles was harvested, centrifuged at 3500 rpm for 5 min and stored at −80 °C until use. 

For HEV, capped genomic HEV RNAs were prepared with the mMESSAGE mMACHINE kit (ThermoFisher Scientific, Waltham, MA, USA) and delivered to PLC3 cells (PLC3/HEV) by electroporation using a Gene Pulser Xcell apparatus (Bio-Rad, Hercules, CA, USA) [20]. Electroporated cells were cultured in HEV medium [20]. At 10 days post-electroporation, cell supernatants (containing eHEV) were collected and stored at −80 °C until use. For the preparation of neHEV, confluent T75 flasks of PLC3/HEV cells were trypsinized, and cells were centrifuged for 10 min at 1500 rpm. Cells were washed thrice with PBS. Intracellular viral particles (neHEV) were extracted by resuspending cells in 1 mL of sterile water at room temperature. Cells were vortexed vigorously for 20 min. and then 110 µL of sterile 10X PBS was added. Samples were clarified by centrifugation 2 min. at 14,000 rpm. The supernatants containing intracellular particles were collected and stored at −80 °C until use. 

### 2.4. Infectious Titers and Quantification of Viral RNA in Infected Cells

Viral inactivation was evaluated by titrating viral titers and by RT-qPCR of viral RNAs in cells inoculated with diluted processed viral preparations. Dilutions starting to 1:100 were used for samples containing milk.

For HCoV-229E infectious titers, Huh-7 cells were seeded in 96-well plates. The following day, cells were infected with serial dilutions of treated/non-treated HCoV-229E samples. Five to six days post-infection, the cytopathic effect was determined in each well to calculate the 50% tissue culture infectious dose (TCID50) titers (TCID50/mL) by using the Spearman and Kärber method.

For HEV infectious titers, Huh-7.5 cells were seeded in 96-well plates. The following day, cells were infected with serial dilutions of treated/non-treated eHEV or neHEV samples. Three days post-infection, cells were fixed and processed for indirect immunofluorescence, as previously described [21]. Cells labeled with the anti-ORF2 antibody 1E6 (Millipore, Antibody registry#AB_827236, Burlington, MA, USA) were counted as infected cells. The number of infected cells was determined for each dilution and used to define the infectious titers in focus forming unit/mL (FFU/mL). 

For quantification of HCoV-229E RNA in infected cells, Huh-7 cells were seeded in 24-well plates. The following day, cells were inoculated with 25 µL of treated/non-treated samples in a final volume of 400 µL of 2% FCS medium for 1 h at 37 °C. The inoculum was replaced with fresh culture medium, and the cells were incubated at 37 °C for 16 h. Cells were lysed using LBP lysis buffer for RNA extraction by following the manufacturer’s instructions (NucleoSpin RNA plus extraction kit; Macherey-Nagel, Düren, Germany). A one-step RT-qPCR reaction was performed using the Takyon Dry Low Rox One-Step RT Probe Mastermix (Eurogentec, Seraing, Belgium) on Quant Studio 3 apparatus (Applied Biosystems, Waltham, MA, USA). Two targets were amplified in multiplex: The HCoV-229E M membrane protein (5′-TTCCGACGTGCTCGAACTTT-3′ (F) and 5′-CCAACACGGTTGTGACAGTGA-3′ (R) and a probe (5′-6FAM-TCCTGAGGTCAATGCA-3′) and the RNaseP (5′-AGATTTGGACCTGCGAGCG-3′ (F) and 5′-GAGCGGCTGTCTCCACAAGT-3′ (R) and a probe (5′-VIC-TTCTGACCTGAAGGCTCTGCG-3′) as an internal control as well as a quality control of extraction method. 

For quantification of HEV RNA in infected cells, Huh-7.5 cells were seeded in 6-well plates. The following day, cells were infected with treated or non-treated HEV particles (m.o.i. = 0.1 for eHEV; m.o.i. = 1 for neHEV). Three days post-infection, HEV RNAs were extracted from cells with the Nucleospin RNA Plus kit (Macherey-Nagel, Düren, Germany). Retrotranscription was performed using the AffinityScript Multiple temperature cDNA synthesis Kit (Agilent Technologies, Santa Clara, CA, USA) according to the manufacturer’s instructions. HEV RNA levels were quantified by using primers (5′-GGTGGTTTCTGGGGTGAC-3′ (F) and 5′-AGGGGTTGGTTGGATGAA-3′ (R)) and a probe (5′-FAM-TGATTCTCAGCCCTTCGC-TAMRA-3′) that target a conserved 70 bp region in the ORF2/3 overlap. Amplifications were done with a Quant Studio 3 apparatus (Applied Biosystems, Waltham, MA, USA) and Taqman universal master mix no AmpErase UNA (Applied Biosystems). Samples with Ct values >35 were considered negative.

### 2.5. Cellular Toxicity Assay

Huh-7 or Huh-7.5 cells were seeded in 96-well plates and incubated for 16 h at 37 °C in a 5% CO_2_ incubator. Cells were then treated with different concentrations of milk diluted in cell culture medium (0 to 1:1000 dilution). Cells were incubated at 37 °C and 5% CO_2_ for 24 h (Huh-7 cells) or 72 h (Huh-7.5 cells). An MTS [3-(4,5-dimethylthiazol-2-yl)-5-(3-carboxymethoxyphenyl)-2-(4-sulfophenyl)-2H-tetrazolium]-based viability assay (Cell Titer 96 aqueous nonradioactive cell proliferation assay; Promega) was performed as recommended by the manufacturer. The absorbance of formazan at 490 nm was detected using a plate reader (BioTek ELx808, Agilent Technologies, Santa Clara, CA, USA). Each measure was performed in triplicate, and each experiment was repeated at least 3 times.

### 2.6. Statistical Analysis

Statistical analyses were performed with GraphPad Prism 9.0. A test was declared statistically significant for any *p* value below 0.05. The non-parametric Kruskal–Wallis test was used. * *p* < 0.05, ** *p* < 0.01, *** *p* < 0.001, **** *p* < 0.0001.

## 3. Results

### 3.1. HHP and HoP Processing of Human Milk Infected with HCoV-229E, eHEV, or neHEV Viruses 

To assess the efficacy of HHP processing and HoP treatment on inactivation of viruses present in human milk, pools of human milk were artificially infected with cell culture-generated viral particles. HCoV-229E particles were used as an enveloped virus model, HEV particles from cell culture supernatant were used as a quasi-enveloped virus (eHEV) model, and HEV particles from cell lysates were used as a non-enveloped virus (neHEV) model [22]. Viruses diluted in cell-culture medium and undiluted viruses were used as controls (Figure 1). Four aliquots of viral samples were treated as follows: One aliquot underwent no treatment (CTL), one aliquot was subjected to 4 cycles of 5 min at 350 MPa at 38 °C (herein designated as 4xHP350 treatment), one aliquot was subjected to 1 cycle of 5 min at 600 MPa at 20 °C (herein designated as 1xHP600 treatment), the last aliquot was subjected to Holder pasteurization (HoP) (Figure 1). After HHP and HoP processing, viral inactivation was evaluated by quantification of infectious particles and of viral RNAs in cells inoculated with processed viral preparations. Of note, before quantifying viral inactivation, we assessed the cellular toxicity of human milk by incubating cells with different dilutions of milk. An MTS assay was performed 24 or 72 h post-inoculation. As shown in Figure 2, dilutions below 1:100 were characterized by cell toxicity. Therefore, dilutions of 1:100 were used for viral titration assays of samples containing milk. 

### 3.2. HHP and HoP Processing of Human Milk Inoculated with HCoV-229E 

Undiluted and medium-diluted HCoV-229E preparations showed a significant decrease in infectivity upon HHP treatment. Indeed, as compared to the control treatment, mean infectivity of undiluted HCoV-229E was decreased by about 4.3 log10 TCID50/mL after 4xHP350 treatment and by about 2.3 log10 TCID50/mL after 1xHP600 treatment (Figure 3A, 229E). Mean infectivity of medium-diluted HCoV-229E was decreased by about 5.3 log10 TCID50/mL after 4xHP350 treatment and by about 2.6 log10 TCID50/mL after 1xHP600 treatment (Figure 3A, 229E/Medium). In contrast, milk-diluted HCoV-229E showed only a 1.4 log10 TCID50/mL decrease in infectivity upon 4xHP350 treatment and no significant reduction in infectivity after 1xHP600 treatment (Figure 3A, 229E/Milk). We also evaluated viral inactivation by quantifying HCoV-229E RNAs in cells inoculated with processed viral preparations (Figure 3B). Undiluted and medium-diluted HCoV-229E preparations showed 3 and 2 log10 reductions of RNA copies/10^6^ cells after 4xHP350 and 1xHP600 treatment, respectively. Milk-diluted HCoV-229E preparations showed 1 log10 reduction of RNA copies/10^6^ cells after 4xHP350 treatment and no reduction of RNAs levels after 1xHP600 treatment (Figure 3B). Together these results indicate that HcoV-229E is inactivated by HHP treatment. In particular, 4xHP350 protocol is more effective in inactivating HcoV-229E. Importantly, our results demonstrate a matrix effect of human milk on HCoV-229E inactivation. 

As recently described for SARS-CoV-2 [23,24], HCoV-229E was significantly inactivated upon HoP treatment, but inactivation was less efficient in milk matrix (Figure 3C,D). Indeed, as compared to the control treatment, mean infectivity of HCoV-229E was decreased by 6 log10 TCID50/mL and 3 log10 reduction of RNA copies/10^6^ cells after HoP treatment, whereas milk-diluted HCoV-229E showed a decrease in infectivity of 1.5 log10 TCID50/mL and 2 log10 reduction of RNA copies/10^6^ cells (Figure 3C,D).

Together these results indicate that HCoV-229E is inactivated by HHP and HoP treatment, but the complex nutritional matrix of DM protects HCoV-229E from inactivation.

### 3.3. HHP and HoP Processing of Human Milk Inoculated with eHEV 

All eHEV preparations showed a significant decrease in infectivity upon 1xHP600 treatment. Indeed, as compared to CTL treatment, mean infectivity was significantly reduced by approximately 3.7 log10 FFU/mL, 2.9 log10 FFU/mL, and 1.6 log10 FFU/mL for undiluted, medium-diluted, and milk-diluted eHEV preparations, respectively, after 1xHP600 treatment (Figure 4A). Surprisingly, infectivity of eHEV samples was not affected by 4xHP350 treatment (Figure 4A). Quantification of HEV RNAs in cells inoculated with treated viral preparations was consistent with these observations (Figure 4B) and showed that 4xHP350 treatment did not reduce eHEV infectivity, whereas 1xHP600 treatment led to a virus inactivation similar to that obtained after 20 min heating at 80 °C, our gold standard procedure for inactivating HEV. In contrast to what was observed for HCoV-229E, we did not observe any matrix effect of human milk on eHEV inactivation by HHP at a pressure of 600 Mpa. Interestingly, whereas HoP treatment fully inactivated undiluted and medium-diluted eHEV samples, no significant viral inactivation was observed for milk-diluted eHEV, indicating a protective effect of milk against inactivation by HoP (Figure 4C,D).

Taken together, our results indicate that eHEV is stable against HHP processing using moderate pressure, and the milk matrix does not protect eHEV from inactivation by HHP processing using a high pressure of 600 MPa. Importantly, the complex nutritional matrix of DM protects eHEV from inactivation by HoP.

### 3.4. HHP and HoP Processing of Human Milk Inoculated with neHEV

As previously observed for eHEV, treatment of neHEV samples by HHP at 600 MPa led to a significant decrease in infectivity. As compared to CTL treatment, mean infectivity was reduced by approximately 4 log10 FFU/mL, 3.4 log10 FFU/mL and 2.7 log10 FFU/mL for undiluted, medium-diluted, and milk-diluted neHEV preparations, respectively (Figure 5A). Quantification of HEV RNAs in cells inoculated with treated viral preparations showed that 1xHP600 treatment led to a 3.7 log10 reduction of RNA copies/µg for neHEV and 6 log10 reduction of RNA copies/µg for medium- and milk-diluted neHEV preparations (Figure 5B). Again, infectivity of neHEV samples was not affected at all by the moderate 4xHP350 treatment (Figure 5A,B). In addition, whereas HoP treatment significantly inactivated undiluted and medium-diluted neHEV samples, no significant viral inactivation was observed for milk-diluted neHEV, indicating again a protective effect of milk against inactivation by HoP (Figure 5C,D).

These results confirm that during HHP treatment, very high pressure must be applied to inactivate HEV, and human milk protects neHEV from inactivation by HoP.

## 4. Discussion

In HMBs, the use of HoP is currently recommended to ensure the microbial safety of DM provided mainly to preterm infants in hospitals [1]. Indeed, some viruses may be present in human milk as a result of secretion from the mammary tissue, notably, cytomegalovirus, human T-lymphocytic virus (HTLV), and human immunodeficiency virus (HIV), or may be present as a contaminant from skin or respiratory droplets either in donor milk or on collection containers used by donor mothers and in HMBs. The safety effect of HoP on inactivation of some viruses in DM and other matrices was recently reviewed by Pitino et al. [25]. Globally, the thermal effect of HoP was shown to be very effective in reducing detectable live viruses for numerous viral types, including cytomegalovirus, HTLV, HIV-1, Ebola, Marburg, and Zika viruses [25]. However, pasteurization degrades numerous heat-sensitive bioactive factors and hormones [1,2,3]. Recently, it has been demonstrated that HHP, which preserves most of the biological properties of human milk, might be an attractive method to sterilize DM [5,6,7,8]. However, the effect of HHP on viruses that may be present in the complex nutritional matrix of human milk has not yet been defined. Here, we explored the capacity of two protocols of HHP to inactivate viruses inoculated to pools of milk. We also used HoP as a gold standard control. Our data indicate that the type of viruses, the nature of their envelope, the biological matrix (culture medium or DM), and the type of treatment (pressure level, number of HHP cycles, temperature during the HHP treatment) are factors of great variability for the efficiency of these viral inactivation methods.

In our study, we used HCoV-229E and HEV to artificially infect DM. Detection of seasonal low-pathogenic HCoV-229E in DM is not documented, and data on whether SARS-CoV-2 could be transmitted from COVID-19 positive mother to the newborn through breastmilk are very limited. Although some case reports described the SARS-CoV-2 detection in breastmilk and infants were diagnosed with COVID-19 (reviewed in [26]), it remains unclear whether the disease is transmitted through breastmilk, direct contact, or through delivery. Regarding HEV, which is found in animal milk [27], only one study reported the isolation of HEV from breast milk during infection [28]. Although contamination of breast milk by coronaviruses and HEV is not well established, they represent good surrogate models for enveloped and non-enveloped viruses.

The first aim of our study was to investigate the effect of HHP processing on viral inactivation in DM. In several recent studies from our group and others, it has been demonstrated that a specific HHP protocol, based on four cycles of a moderate pressure (350 MPa) of 5 min each performed at 38 °C [6] (herein designated as 4xHP350 treatment), sterilizes DM but also maintains bioactive factors, such as immunoglobulins, lactoferrin, lysozyme, BSSL enzyme, oligosaccharides, and hormones at levels close to raw milk [5,6,7,8]. Thus, we used this specific HHP protocol to evaluate its effect on the survival of our surrogate viruses, i.e., HCoV-229E, eHEV, and neHEV, diluted in culture medium and DM. Interestingly, we found that the 4xHP350 protocol drastically inactivates HCoV-229E when undiluted or diluted in medium. However, by treating infected milk-diluted samples, we found that living infectious HCoV-229E particles remain after treatment, indicating a matrix-protecting effect of human milk on viral inactivation. Our data are the first to demonstrate that coronavirus infectivity is reduced by high-pressure treatment. However, our results also indicate that the 4xHP350 protocol is likely not suitable to inactivate Coronaviruses and likely more generally enveloped viruses present in breast milk. In addition, we showed that the 4xHP350 protocol was ineffective on eHEV and neHEV infectivity in both culture medium and DM, indicating that this process is likely not suitable for inactivating quasi-enveloped and non-enveloped viruses. Thus, the 4xHP350 protocol, which preserves many nutritional and bioactive factors, is not suitable for ensuring the viral safety of donor milk.

To investigate if a higher pressure was more effective against our model viruses, we treated samples with a higher pressure of 600 MPa at 20 °C for 5 min (herein designated as 1xHP600 treatment). We observed that this single strong HHP treatment was less effective than 4xHP350 treatment in HCoV-229E inactivation and inefficient for samples diluted in milk. In contrast, the 1xHP600 treatment highly inactivated HEV particles, as observed in previous studies [29,30]. Moreover, we found no matrix effect of human milk on HEV inactivation by the 1xHP600 protocol, which contrasts with another study using artificially contaminated pork pate [29]. Hence, our study provides new evidence that pressure and temperature processing parameters and the biological matrix influence the efficacy of HHP treatment in viral inactivation.

Finally, we analyzed viral inactivation upon HoP. Recently, it has been shown that HoP of human milk inactivates SARS-CoV-2 [23,24]. Here, we compared the effect of HoP on the inactivation of HCoV-229E undiluted or diluted in DM. Although HoP effectively reduced infectivity of both viral preparations, inactivation was less efficient in milk matrix. In addition, we demonstrated that HoP, although effective in inactivating undiluted and medium-diluted HEV particles, does not significantly inactivate HEV in DM. Together, these data highlight the importance of taking into account the biological matrix in viral inactivation assays. In accordance with our results, Huang et al. demonstrated that gavage with HEV-contaminated raw and pasteurized cow milk resulted in an active infection in rhesus macaques and that a short period of boiling, but not pasteurization, was able to completely inactivate HEV [31]. Our data are also consistent with previous findings demonstrating that, to completely inactivate non-enveloped viruses, such as hepatitis A or porcine parvovirus, in human milk or in other matrices, temperatures above 63 °C (usually tested in studies between 70 and 90 °C) or a significantly longer duration (more than 30 min) at 60–63 °C, are generally required [25]. Hence, a higher thermal treatment of DM may be used instead of the actual HoP procedure performed at 62.5 °C in HMBs.

## 5. Conclusions

In conclusion, we demonstrated that HHP and HoP treatments cannot guarantee total viral safety of donor milk artificially infected with two surrogate models for enveloped and non-enveloped viruses. Our study emphasizes the critical importance of viral testing in HMBs.

## Figures and Tables

**Figure 1 viruses-15-01571-f001:**
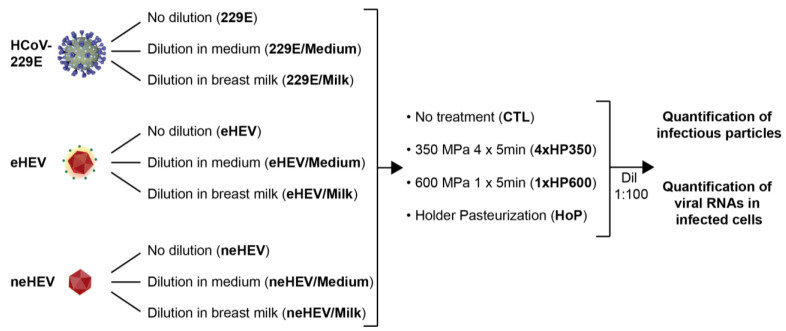
Methodology used to assess the efficacy of HHP and HoP treatments on inactivation of viruses in human milk. HCoV-229E, enveloped HEV (eHEV) and non-enveloped HEV (neHEV) viruses were kept undiluted or diluted (1:2) in cell-culture medium or in breast milk. Four aliquots of viral samples were treated as follows: One aliquot underwent no treatment (CTL), one aliquot was subjected to 4 cycles of 5 min at 350 MPa (4xHP350 treatment) at 38 °C, one aliquot was subjected to 1 cycle of 5 min at 600 MPa (1xHP600 treatment) at 20 °C, the last aliquot was subjected to Holder pasteurization (HoP; 30 min at 62.5 °C). Viral inactivation was evaluated by titrating viral titers (quantification of infectious particles) and by RT-qPCR of viral RNAs in cells inoculated with processed viral preparations. Dilutions of 1:100 (Dil 1:100) were used for viral titration assays of samples containing milk.

**Figure 2 viruses-15-01571-f002:**
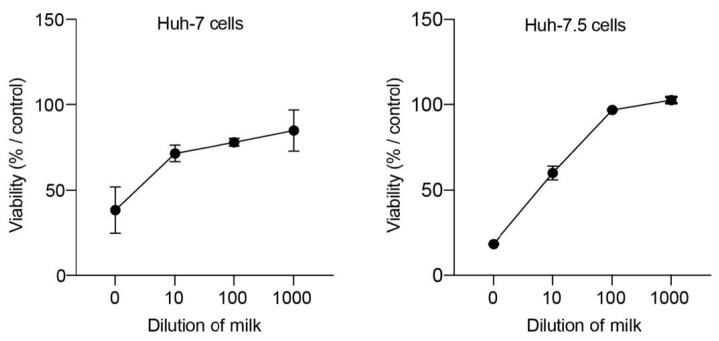
Dose-response curves of Huh-7 and Huh-7.5 cells incubated with human milk. Cells were cultured in the presence of different dilutions of milk (pur to 1:1000 dilution) for 24 h (Huh-7 cells) or 72 h (Huh-7.5 cells). Cell viability was determined by an MTS-based assay. Cells cultured in the absence of milk were used as a control and set to 100%.

**Figure 3 viruses-15-01571-f003:**
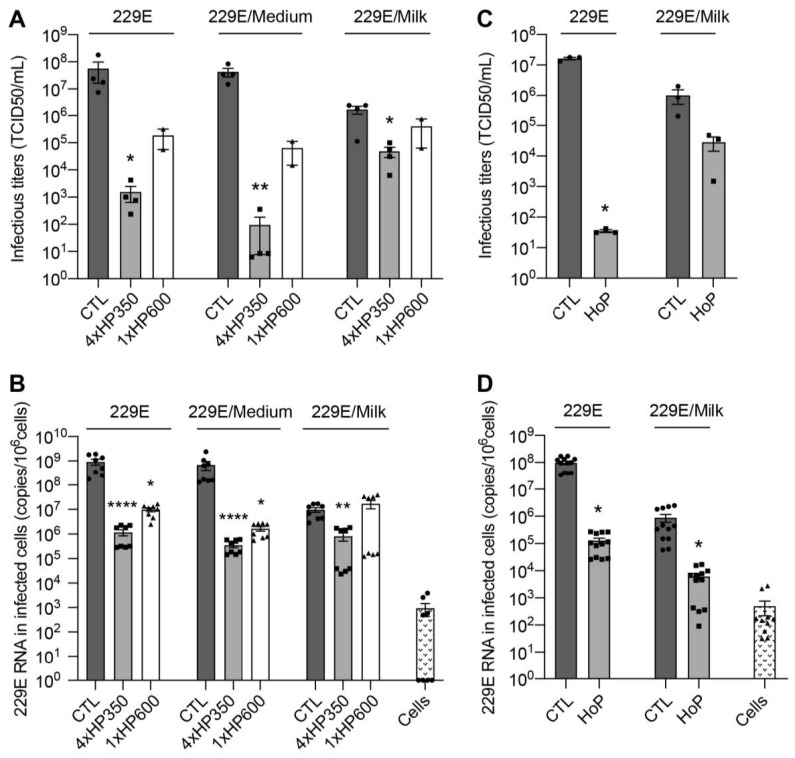
HHP and HoP processing of human milk inoculated with HcoV-229E. (**A**,**B**) Undiluted (229E), medium-diluted HcoV-229E (229E/Medium), and milk-diluted HCoV-229E (229E/Milk) preparations were left untreated (CTL), treated with 4 cycles of 5 min at 350 MPa (4xHP350 treatment) or treated with 1 cycle of 5 min at 600 MPa (1xHP600 treatment). (**C**,**D**) Undiluted (229E) and milk-diluted HCoV-229E (229E/Milk) preparations were left untreated (CTL) or treated by Holder pasteurization (HoP). Infectivity of treated viral preparations was evaluated by measuring infectious titers (**A**,**C**) and by quantifying viral RNA in Huh-7 cells inoculated with treated samples (**B**,**D**). mean ± S.E.M., Kruskal–Wallis test, *****
*p* < 0.05, ******
*p* < 0.01, ********
*p* < 0.0001.

**Figure 4 viruses-15-01571-f004:**
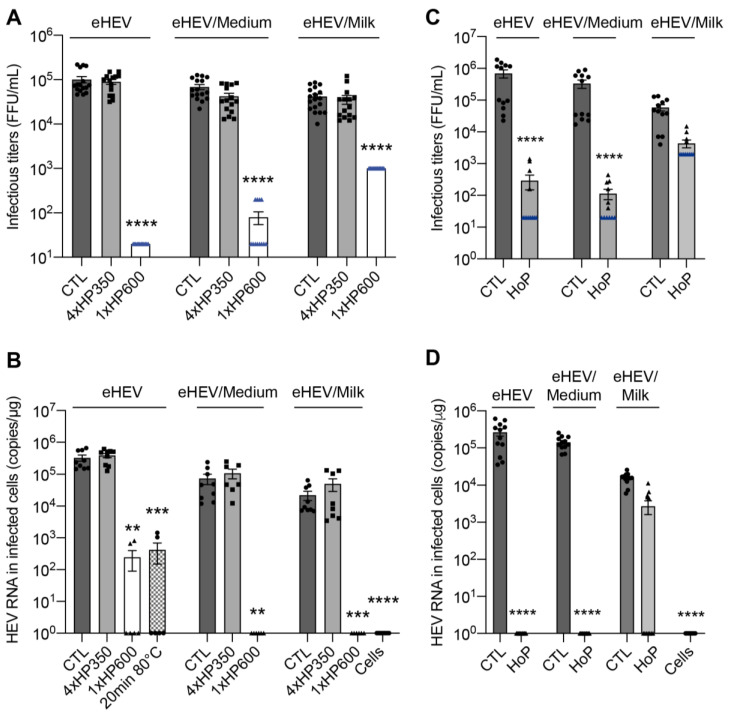
HHP and HoP processing of human milk inoculated with eHEV. (**A**,**B**) Undiluted (eHEV), medium-diluted eHEV (eHEV/Medium), and milk-diluted eHEV (eHEV/Milk) preparations were left untreated (CTL), treated with 4 cycles of 5 min at 350 MPa (4xHP350 treatment) or treated with 1 cycle of 5 min at 600 MPa (1xHP600 treatment). (**C**,**D**) eHEV preparations were left untreated (CTL) or treated by Holder pasteurization (HoP). Infectivity of treated viral preparations was evaluated by measuring infectious titers (**A**,**C**) and by quantifying viral RNA in Huh-7.5 cells inoculated with treated samples (**B**,**D**). In blue are the detection thresholds. mean ± S.E.M., Kruskal–Wallis test, ******
*p* < 0.01, *******
*p* < 0.001, ********
*p* < 0.0001.

**Figure 5 viruses-15-01571-f005:**
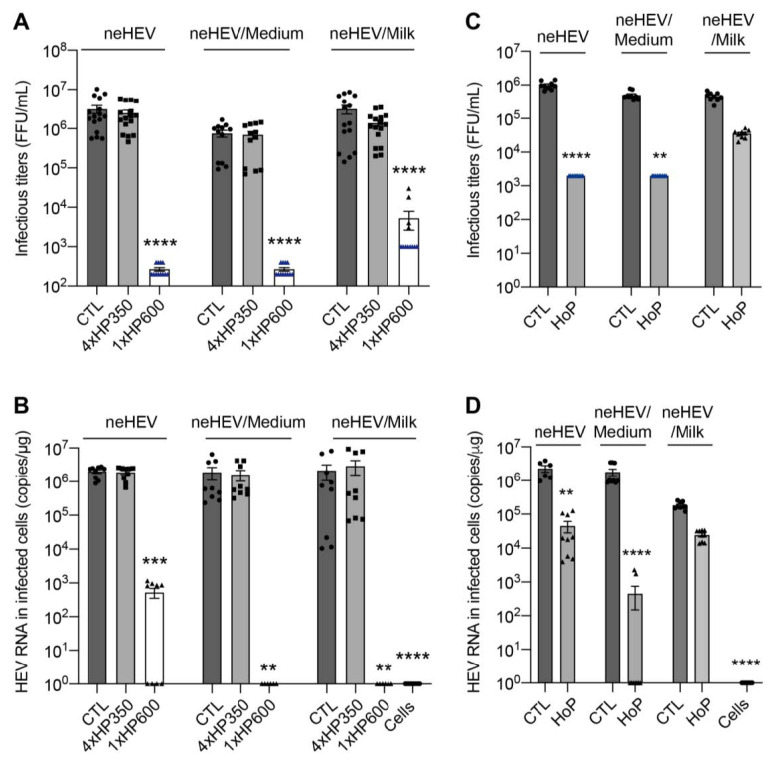
HHP and HoP processing of human milk inoculated with neHEV. (**A**,**B**) Undiluted (neHEV), medium-diluted neHEV (neHEV/Medium), and milk-diluted neHEV (neHEV/Milk) preparations were left untreated (CTL), treated with 4 cycles of 5 min at 350 MPa (4xHP350 treatment), or treated with 1 cycle of 5 min at 600 MPa (1xHP600 treatment). (**C**,**D**) neHEV preparations were left untreated (CTL) or treated by Holder pasteurization (HoP). Infectivity of treated viral preparations was evaluated by measuring infectious titers (**A**,**C**) and by quantifying viral RNA in Huh-7.5 cells inoculated with treated samples (**B**,**D**). In blue are the detection thresholds. mean ± S.E.M., Kruskal–Wallis test, ******
*p* < 0.01, *******
*p* < 0.001, ********
*p* < 0.0001.

## Data Availability

The datasets generated and analyzed during the current study are available from the corresponding author on reasonable request.

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
