# Peer review of "Effect of High Hydrostatic Pressure Processing and Holder Pasteurization of Human Milk on Inactivation of Human Coronavirus 229E and Hepatitis E Virus"

_viruses, 2023, doi:10.3390/v15071571_

Round 1

Reviewer 1 Report

In this study, Bouquet et al. investigated the effect of high hydrostatic pressure processing on milk infected with 229E and HEV. The results are clearly presented and the virological data solid, the translational relevance and aim of the study is not clear. Is the study important for the food industry, for breastfeeding or a virus inactivation study?

Specific comments to improve the study are the following:

-What is the relevance of the study, this should be clearly stated in the introduction and aim.

-Are coronaviruses detected in human milk? The testing of 229E is not clear or is corona a surrogate virus for envelop viruses?

-The data in Figure 2 show that at least a 1:10 is needed to exclude cytotoxicity. However, in Figure 1 a 1/2 diltution is stated? The material and method section are msiing all that information to understand the results section of Figure 3 and following.

-The study by Huang et al. could not be confirmed in the field. So, there is no risk for HEV in cow milk:

https://aasldpubs.onlinelibrary.wiley.com/doi/pdf/10.1002/hep.28863

Please also explain here the relevance to test HEV in these setting?

Author Response

Please note that we have attached to "non-published material" the revised manuscript with highlighted changes.

Reviewer 2 Report

Sterilized donor milk (DM) is frequently used for feeding when breast 17 milk is lacking. The authors managed to detect several methods of the efficiency on inactivate coronavirus and HEV. The paper is overall interesting.

My only concern is that the clinical and public health relevance of the chance of these viruses' presentation in the milk product.

The paper lack of the basic information about the prevalence studies of the selected viruses presented in human milk. In the opinion of this reviewer, the evidence that support HEV can shed in human milk is thin. There is only one case report demonstrated HEV RNA presentation in an acute hepatitis E patient. Further studies mainly investigated the presentation of HEV in animals' milks but the evidence is contradictory.

I think the rationale of this study should be further specified. What compelling evidence urges the need of the development of inactivation techniques against these two viruses in human milk.

Author Response

(The authors gave the same response as above.)

Round 2

Reviewer 1 Report

All points addressed.

no comment